# Soil Carbon Storage, Enzymatic Stoichiometry, and Ecosystem Functions in Indian Himalayan Legume-Diversified Pastures

Avijit Ghosh [1,*], Suheel Ahmad [2], Amit K. Singh [1], Pramod Jha [3], Rajendra Kumar Yadav [4], Raimundo Jiménez Ballesta [5,*], Sheeraz Saleem Bhatt [2], Nagaratna Biradar [6] and Nazim Hamid Mir [2]

1   ICAR-IGFRI, Jhansi 284 003, India; amit09bhu@gmail.com
2   Hilly Regional Research Station, ICAR-IGFRI, Srinagar 190 005, India; suhail114@gmail.com (S.A.);
    shrzbhat@gmail.com (S.S.B.); nazimmir.nazu9@gmail.com (N.H.M.)
3   ICAR-IISS, Bhopal 462 038, India; jha_iari@yahoo.com
4   Department of Soil Science, Agriculture University, Kota 324 001, India; raj91yadav@gmail.com
5   Department of Geology and Geochemistry, Autónoma University of Madrid, 28049 Madrid, Spain
6   Southern Regional Research Station, ICAR-IGFRI, Dharwad 580 005, India; nagaratna123@gmail.com
*   Correspondence: avijit.ghosh@icar.gov.in (A.G.); raimundo.jimenez@uam.es (R.J.B.)

**Abstract:** The influences of legume diversification on soil carbon (C) pools and sequestration, particularly in Himalayan pastureland, remain poorly understood. Moreover, the impact of legume diversification activities and the stoichiometry of soil enzymes in C biogeochemistry at the ecosystem level remains largely overlooked. The purpose of this study is to investigate the influences of legume diversification on activities and the stoichiometry of soil enzymes and their control of C sequestration in pasturelands. Four experimental fertilized species combinations, namely, SG (50% *Festuca arundinacea* + 50% *Dactylis glomerata*), SGL1 (25% *Festuca arundinacea* + 25% *Dactylis glomerata* + 50% *Onobrychis viciifolia*), SGL2 (25% *Festuca arundinacea* + 25% *Dactylis glomerata* + 50% *Trifolium pratense*), SGL12 (25% *Festuca arundinacea* + 25% *Dactylis glomerata* + 25% *Onobrychis viciifolia* + 25% *Trifolium pratense*), and natural pasture (NG) were compared. Soils under SGL1, SGL2, and SG12 had ~18, 36, and 22% greater soil C than SG, respectively. Among the pastures with fertilization, the C mineralization was suppressed by legume diversification. C sequestration under SGL1, SGL2, and SG12 was ~27, 22, and 38% higher than SG, respectively, at the 0–30 cm soil layer. The ratios of DHA are as follows: for PhOX and DHA, PerOX significantly decreased with an increasing grass–legume mixture, suggesting greater C sequestration. PCA analysis revealed that C sequestration under legume diversification and enzymatic stoichiometry had an indirect but substantial impact on C sequestration. The increasing C sequestration under SGL12 was complemented by higher productivity. Data suggested that increasing legumes in pastureland might greatly enhance ecosystem functions such as soil C storage, productivity, ecorestoration efficiency, and biological activity in Indian Himalayan pastureland.

**Keywords:** temperate pasture; soil organic carbon; microbial functions; legume diversification; ecosystem services



## 1. Introduction

Because assessments indicate that global carbon reserves in pastures might be ~50% larger than forests [1], pastures have a more significant capability to sequester $CO_2$ than agricultural regions [2]. However, estimations showed that between 50% and 70% of all pastureland worldwide is deteriorated, only able to maintain one animal unit (1 unit = 450 kg of an animal's body weight) per hectare [3]. In an attempt to optimize this situation, legume diversification was used for pasture improvement, with the goals of decreasing deteriorated pastures [4], promoting financial profits [5], limiting net $CO_2$ pollution [6], and optimizing soil properties, particularly the quantity of soil organic carbon (SOC) [7]. In comparison to natural pastures, the managed pastures might be more prolific with a larger carrying

capability. Compared to native pastures, managed pasture species frequently offer greater feed quality and longer growth periods. More livestock may acquire feed for a longer period of time if the quality and quantity of feed are increased. Many investigations show that modifications in pasture diversity, especially in temperate grasslands, might alter the responsiveness of plant production and, consequently, ecosystem C supplies. Nevertheless, because of the intricate link between the soil microbial population and the dynamics of soil organic carbon, the belowground C cycling under grass–legume diversification is not yet comprehensively interpreted. They control several important biogeochemical mechanisms and ecosystem functions, along with soil C cycling and soil enzymes in the terrestrial ecosystem [5]. To forecast how the soil C pool on pastureland will react to potential climatic changes, it is thus necessary to better understand facts regarding the change in the soil enzyme activity and stoichiometry and their connections with the change in SOC.

Some characteristics, such as enzymatic activity and stoichiometry, have a significant impact on the redistribution, stability, and mineralization of carbon in soil [6,7]. Indications of the distribution and sequestration of C in soil could be found in stoichiometric ratios of soil enzymes such as dehydrogenase (DHA), FDA, urease (URE), alkaline phosphatase (ALP), phenol oxidase (PhOX), and peroxidase (PerOX). Using data derived from 40 ecosystems, a worldwide meta-analysis revealed that ratios of soil-specific C, N, and P, hydrolyzing enzymatic activities, are very near to 1:1:1 [8]. Those investigations were expanded, and the researchers discovered that the C, N, and P enzymatic activities in terrestrial soils had identical stoichiometry ratios [9]. On sown pasture fields in temperate regions, it is uncertain how changes in enzymatic stoichiometry potentially affect C sequestration.

The characteristics that regulate C sequestration are considerably changed by modifications in the pastureland's relative grass–legume mixture. It is generally known that plant community diversification has favorable impacts on soil functions and production [10]. Diversification can have a substantial impact on pasture production and soil C storage with the introduction of N-fixing legumes [11,12]. Legumes can also increase soil biodiversity and change ecological processes [13]. Yet, there is also a noted rise in pasture production without legumes, which was ascribed to better nutrient uptake and use efficiency [14]. Several research works were looking at how changes in grass–legumes affect soil C, enzyme activity, and stoichiometry [15,16]. Previous studies revealed legumes as a major contributor to primary production, carbon sequestration, microbiological diversity, and activity [17,18]. Microbial richness, C mineralization, and plant production can all be impacted by the distribution of C in its many forms [19,20]. In order to promote C storage and avoid C loss from temperate pastureland, we hypothesized that increasing legume variety would change microbial activity and enzyme stoichiometry.

The impacts of legumes on soil fertility have been the subject of certain research to date [21,22]. The impact of grass–legume mixture on native pastureland was little known, but these research works shed light on some of the parameters influencing C storage in soil. Although this knowledge would be crucial for maintaining improved pastureland, there have been surprisingly few studies that examined the impact of grass–legume diversity on the dynamics and sequestration of soil C in temperate pastureland. Consequently, the objectives of this work were to (i) evaluate the impacts of growing legumes on soil C distribution and enzyme activity and (ii) quantify the impact of enzyme stoichiometry on soil C sequestration following legume diversification in temperate grassland.

## 2. Materials and Methods

### 2.1. Description of Research Site and Experiment

The experiment was conducted in the research farm of Himalayan Fodder and Grassland Research Station, Srinagar of ICAR-Indian Grassland and Fodder Research Institute ($34°46'$ N latitude and $74°47'$ E longitude and 1630 m above mean sea level) (Figure 1). The climate was temperate with mild hot summers whereas winters were generally wet, frozen with moderate to heavy snowfall. Dry spells varying in intensity and frequency were observed from late summer to autumn. The maximum temperature observed during

the experimental period (2016–2022) was as high as 35 °C during July and August months while sub-freezing temperatures and frost were common during winters (−11 °C). Climatically, the site is located in mid altitude temperate zone, which is characterized by hot summers and severe winters. The average annual precipitation during the experimental period (2016–2022) is about 700 mm, and more than 80 percent of precipitation is received from western disturbances.

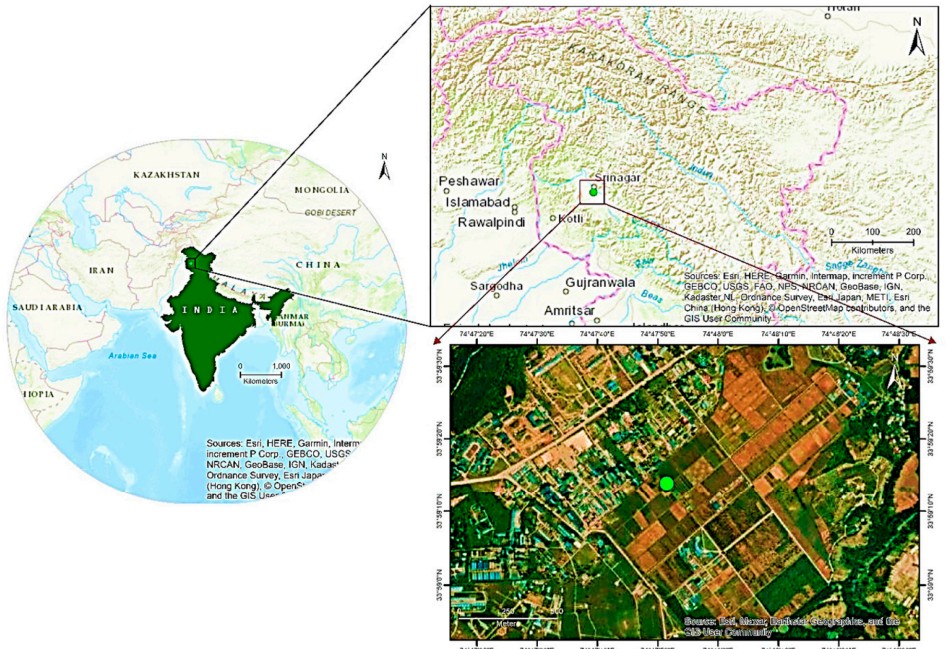

| SG | SGL12 | SGL2 |
| --- | --- | --- |
| SGL1 | NG | SG |
| SGL2 | SG | SGL1 |
| SGL12 | SGL2 | SGL12 |
| NG | SGL1 | NG |
| F | F | F |

**Figure 1.** Geometric location and treatment distribution in randomized block design of the study area in the Indian Himalayan region.

The soil had silt loam texture (26% sand, 54% silt, and 20% clay), near-to-neutral pH (6.70), and non-saline reaction (0.08 ds m$^{-1}$). The soil was low in total organic carbon (13,455 kg ha$^{-1}$), nitrogen (280 kg ha$^{-1}$), phosphorus (17 kg ha$^{-1}$), and high in potassium (468 kg ha$^{-1}$) prior to experimentation. These soils are classified as Typic Udorthents in the Mesic temperature regime. The investigation involved grasses (*Festuca arundinacea* and *Dactylis glomerata*) and legumes (*Onobrychis viciifolia* and *Trifolium pratense*) for studying the impact of legume diversification on soil C sequestration, C pools, and soil enzyme activity. Five pasture combinations have been considered for the experiment since 2016, namely, SG (50% *Festuca arundinacea* + 50% *Dactylis glomerata*), SGL1 (25% *Festuca arundinacea* + 25% *Dactylis glomerata* + 50% *Onobrychis viciifolia*), SGL2 (25% *Festuca arundinacea* + 25% *Dactylis glomerata* + 50% *Trifolium pratense*), SGL12 (25% *Festuca arundinacea* + 25% *Dactylis glomerata* + 25% *Onobrychis viciifolia* + 25% *Trifolium pratense*), and NG (natural pasture) and F (uncultivated fallow). In the study region, local farmers commonly use natural pasture as source of fodder for animals. We aimed to understand the impact of legume diversification with recommended input supply, including irrigation and fertilizer. Hence, we compared the results with natural pasture (without fertilizer application) and fallow (without crop). The natural pastures comprised *Festuca arundinacea*, *Dactylis glomerata*, *Bromus* spp., *Phalaris* spp., *Brachiaria* spp., *Paspalam* spp., *Chrysopogon* spp., *Bothriocloa* spp., *Setaria* spp., *Chenopodium glaucum*, *Capsella bursa-pastoris*, *Plantogo lanceolata*, *Daucus carota*, *Convolvulus arvensis*, and low-yielding grasses (*Poa* spp., *Agropyron* spp., *Bromus* spp., *Alopecurus* spp., *Avena fatua*, etc.) and shrubs like *Rubus* spp. and *Zizyphus* spp. There are no legume species in NG. The uncultivated fallow does not contain any crops or grass. Pastures were sown in 2016. Seed rates of 15 kg ha$^{-1}$ for *Festuca arundinacea* and *Dactylis glomerata*, 8 kg ha$^{-1}$ for *Onobrychis viciifolia,* and 35 kg ha$^{-1}$ for *Trifolium pratense*

were maintained. The population of grasses and legumes was maintained throughout the experimentation period by transplanting or replanting as and where needed. The seeds were sown in lines 30 cm apart. A uniform plot size of 8 m × 8 m was used for investigations. A similar fallow land (with similar slope, topography, soil texture, and parent material) was maintained since the inception of the experiment. The study consisted of six treatments laid out in a randomized block design (RBD) with four replications. The study site was not previously cultivated and was a degraded land with lots of weed flora, including forbs, *Chenopodium glaucum*, *Capsella bursa-pastoris*, *Plantogo lanceolata*, *Daucus carota*, *Convolvulus arvensis*, and low-yielding grasses (*Poa* spp., *Agropyron* spp., *Bromus* spp., *Alopecurus* spp., *Avena fatua*, etc.) and shrubs like *Rubus* spp. and *Zizyphus* spp. The NG was maintained with these florae. Uniform dose of N, $P_2O_5$, and $K_2O$ (80, 40, and 30 kg ha$^{-1}$, respectively) was applied in May every year to SG, SGL1, SGL2, SGL12, as well as F.

### 2.2. Collection of Plant and Soil Samples

The biomass yield of each pasture was estimated after harvesting grasses during May, August, and March of every year. The biomass yield was measured as dry weight of harvests. Soil samples were taken from surface layer (SL: 0–15 cm) and sub-surface layer (SSL: 15–30 cm) during March 2022 using a core sampler. Soil samples were allowed to dry in the sun and pass through a 2 mm sieve before analysis of soil C pools. A subsample was stored at 4 °C for assessing enzyme activities and other microbial parameters. The field-moist soil (60% of WHC) was allowed to equilibrate at room temperature for 24 h prior to analysis.

### 2.3. Soil Organic Carbon Estimation and C Mineralization Study

Total soil organic carbon (TOC) and labile and recalcitrant carbon (LC and RC) were measured using wet oxidation method [23]; particulate and mineral organic matter associated C (POMC and MOMC) were measured using sodium hexametaphosphate as extractant [24]; water-soluble C (WSC) was measured using wet oxidation method [25]; and active C (AC), using potassium permanganate oxidation method [26], was estimated for each soil samples. Carbon mineralization from the soil samples was measured for 45 days using incubation, followed by entrapment method [16]. The decay kinetics was estimated using the first-order decomposition reaction:

$$Ct = Co(1 - \exp(kc \times t)), \tag{1}$$

indicating cumulative C mineralization (*Ct*) with C decay rate (*kc*) after *t* days of incubation at constant temperature.

Bulk density (*BD*) of soil was estimated using a core sampler. The C accumulation (*Cacc*) and sequestration (*Cseq*) were quantified as

$$Cacc = TOC * BD * depth\ of\ soil\ layer \tag{2}$$

$$Cseq = RC * BD * depth\ of\ soil\ layer \tag{3}$$

### 2.4. Soil Enzyme Activity Measurement and Determination of Soil Functional Diversity Indices

The activities of dehydrogenase (DHA), fluorescein di acetate (FDA), alkaline phosphatase (ALP), urase (URE), phenol oxidase (PhOX), and peroxidase (PerOX) were estimated for each soil sample [27]. Microbial biomass C was also measured for each sample [28]. The enzymatic stoichiometry was calculated with respect to DHA activity. For this, enzyme activity was converted to logarithmic values and expressed as DHA: FDA, DHA: URE, DHA: ALP, DHA: PhOX, and DHA: PerOX. For example, the DHA: FDA value of 1.47 indicates that for 1.47 units of DHA activity, 1 unit of FDA activity would happen, implying that the narrower the ratio, the lesser the decomposition of SOM and the better

the stability of C in soil. Soil functional diversity, such as Shanon diversity index (H) and Simpson–Yule Index (SYI), was also calculated [29].

### 2.5. Estimation of Indices for Ecosystem Function

Ecosystem functions such as carbon storage, microbial activity, and soil restoration efficiency were estimated for each pasture system. Carbon management index (CMI) [30], biological activity index (BAI) [31], and ecorestoration efficiency of pastures (ERE) [31] were computed.

### 2.6. Analytical Procedure of Data

The data were subjected to analysis of variance as applicable to RBD using Tukey's test for mean differentiation. The following model was used:

$$yij = \mu + \tau i + bj + \epsilon ij$$

where $\mu$ is the general mean, $\tau i$ are the main effects of treatments, $bj$ are the effects of blocks, and $\epsilon ij$ is a usual independent and identically distributed error term with expectation zero and variance $\sigma^2$.

Pearson Correlation was analyzed among enzyme stoichiometries with C mineralization (Ct), accumulation (Cacc), and sequestration (Cseq). A non-metric multidimensional scaling (NMDS) was performed using a Bray–Curtis similarity matrix to determine the similarities of SOC pools, soil enzyme activity, and stoichiometry among different pastures. A principle component analysis (PCA) was performed, and two most important components were extracted to understand the direct and indirect roles of microbial activity and their stoichiometry on C accumulation and sequestration.

## 3. Result

### 3.1. Soil pH, EC, Bulk Density, and Pasture Productivity

Soils under SG, SGL1, SGL2, SG12, and NG had neutral soil pH and non-saline electrical conductivity. The bulk density of soil did not differ significantly but varied between 1.31 to 1.41 Mg m$^{-3}$. Compared to NG, the productivity of experimental plots is higher, but this can be explained by fertilizer. The biomass production under SG, SGL1, SGL2, and SG12 was ~2.1, 2.5, 2.4, and 2.2 times greater than NG, respectively. However, biomass production under SGL1, SGL2, and SG12 was ~18, 12, and 6% higher than SG, respectively (Table 1).

**Table 1.** The soil pH, electrical conductivity (EC; ds m$^{-1}$), bulk density (Mg m$^{-3}$), and pasture productivity (Mg ha$^{-1}$) in Indian Himalayan fertilized and natural pastures. Values with same small-case letters are statistically similar.

| Parameters | Soil pH | | Soil EC | | Bulk Density | | Productivity |
|---|---|---|---|---|---|---|---|
| | | | Soil Layer | | | | |
| Treatments [#] | 0–15 cm | 15–30 cm | 0–15 cm | 15–30 cm | 0–15 cm | 15–30 cm | |
| SG | 6.49 a | 6.86 a | 0.08 a | 0.04 a | 1.36 a | 1.38 a | 8.98 b |
| SGL1 | 6.87 a | 7.11 a | 0.10 a | 0.05 a | 1.33 a | 1.34 a | 10.59 a |
| SGL2 | 6.32 a | 6.37 a | 0.13 a | 0.12 a | 1.31 a | 1.39 a | 10.02 a |
| SGL12 | 6.78 a | 6.81 a | 0.07 a | 0.08 a | 1.33 a | 1.33 a | 9.48 ab |
| NG | 6.33 a | 6.97 a | 0.09 a | 0.06 a | 1.34 a | 1.36 a | 4.22 c |
| F | 6.54 a | 6.62 a | 0.12 a | 0.11 a | 1.40 a | 1.41 a | --- |

[#] SG (50% *Festuca arundinacea* + 50% *Dactylis glomerata*), SGL1 (25% *Festuca arundinacea* + 25% *Dactylis glomerata* + 50% *Onobrychis viciifolia*), SGL2 (25% *Festuca arundinacea* + 25% *Dactylis glomerata* + 50% *Trifolium pratense*), SGL12 (25% *Festuca arundinacea* + 25% *Dactylis glomerata* + 25% *Onobrychis viciifolia* + 25% *Trifolium pratense*), and NG (natural pasture) and F (uncultivated fallow).

### 3.2. Total Soil Organic Carbon (TOC), Microbial Biomass C (MBC), Labile C (LC) and Recalcitrant C (RC)

The system diversification with legume inclusion significantly impacted the TOC, MBC, LC, and RC. Soils of SL under SGL1, SGL2, and SG12 had ~18, 36, and 22% greater TOC than SG, respectively. The SGL1, SGL2, and SG12 had ~15, 30, and 18% greater MBC than SG, respectively (Table 2). They contained ~12, 26, and 11% higher LC as well as 27, 51, and 38% greater RC, respectively, over SG. At the SSL, SGL1 and SGL12 had ~25 and 39% higher TOC, 20 and 32% higher LC, and 32 and 47% higher RC than SG. However, at the SSL, SG and SGL2 had identical TOC and RC contents, although SG had significantly greater LC than SGL2. At the SL and SSL, NG had significantly greater TOC, MBC, LC, and RC than SG, indicating the detrimental impact of SG on soil properties (Table 2).

**Table 2.** Total organic carbon (TOC; g kg$^{-1}$), labile C (LC; g kg$^{-1}$), recalcitrant C (RC; g kg$^{-1}$), and microbial biomass C (MBC; mg kg$^{-1}$) in Indian Himalayan fertilized and natural pastures. Values with same small-case letters are statistically similar.

| Parameters | TOC | | LC | | RC | | MBC | |
|---|---|---|---|---|---|---|---|---|
| | | | | **Soil Layer** | | | | |
| Treatments [#] | 0–15 cm | 15–30 cm | 0–15 cm | 15–30 cm | 0–15 cm | 15–30 cm | 0–15 cm | 15–30 cm |
| SG | 9.08 b | 7.99 b | 5.42 b | 4.47 b | 3.66 bc | 3.52 c | 257.71 b | 232.24 b |
| SGL1 | 10.71 ab | 9.98 ab | 6.04 a | 5.34 a | 4.67 b | 4.64 b | 295.92 ab | 278.94 a |
| SGL2 | 12.34 a | 7.26 b | 6.82 a | 3.86 c | 5.53 a | 3.41 c | 334.13 a | 215.26 b |
| SGL12 | 11.07 a | 11.07 a | 6.02 a | 5.91 a | 5.05 a | 5.16 a | 304.41 ab | 304.41 a |
| NG | 10.17 ab | 8.90 b | 5.54 b | 4.69 b | 4.63 b | 4.21 b | 283.19 b | 253.47 ab |
| F | 6.90 c | 6.40 c | 3.81 c | 3.42 c | 3.09 c | 2.98 d | 206.71 c | 195.00 c |

[#] SG (50% *Festuca arundinacea* + 50% *Dactylis glomerata*), SGL1 (25% *Festuca arundinacea* + 25% *Dactylis glomerata* + 50% *Onobrychis viciifolia*), SGL2 (25% *Festuca arundinacea* + 25% *Dactylis glomerata* + 50% *Trifolium pratense*), SGL12 (25% *Festuca arundinacea* + 25% *Dactylis glomerata* + 25% *Onobrychis viciifolia* + 25% *Trifolium pratense*), and NG (natural pasture) and F (uncultivated fallow).

### 3.3. Soil Organic Carbon Fractions

The legume diversification strategy did not affect WSC at any soil layer significantly. At the SL, AC content under SGL1, SGL2, and SG12 increased by ~9, 10, and 17% over SG, respectively. The POMC increased significantly only under SGL2 over SG. The MOMC was boosted through SGL1, SGL2, and SG12 by ~13, 23, and 26% over SG, respectively. Interestingly, at the SSL, SGL2 decreased the AC and POMC; however, an increase in MOMC increased by ~33% over SG. The SG and NG had similar concertations of WSC, AC, POMC, and MOMC at both soil layers (Table 3).

**Table 3.** Water-soluble carbon (WSC; mg kg$^{-1}$), active C (AC; g kg$^{-1}$), particulate organic matter associated C (POMC; g kg$^{-1}$), mineral organic matter associated C (MOMC; g kg$^{-1}$), and C decay rate (kc, (day$^{-1}$)) in Indian Himalayan fertilized and natural pastures. Values with same small-case letters are statistically similar.

| Parameters | WSC | | AC | | POM-C | | MOM-C | | Decay Rate | |
|---|---|---|---|---|---|---|---|---|---|---|
| | | | | | **Soil Layer** | | | | | |
| Treatments [#] | 0–15 cm | 15–30 cm | 0–15 cm | 15–30 cm | 0–15 cm | 15–30 cm | 0–15 cm | 15–30 cm | 0–15 cm | 15–30 cm |
| SG | 247.99 a | 233.04 a | 1.04 c | 0.98 bc | 2.29 c | 2.09 cd | 3.62 cd | 3.57 cd | $2.40 \times 10^{-2}$ a | $6.81 \times 10^{-4}$ c |
| SGL1 | 257.82 a | 246.84 a | 1.13 b | 1.09 ab | 2.43 b | 2.28 b | 4.10 b | 4.09 b | $2.06 \times 10^{-3}$ b | $5.63 \times 10^{-4}$ d |
| SGL2 | 269.99 a | 223.46 a | 1.22 a | 0.94 c | 2.59 a | 1.96 d | 4.46 a | 4.73 a | $1.47 \times 10^{-4}$ d | $3.99 \times 10^{-4}$ e |
| SGL12 | 257.47 a | 255.73 a | 1.15 ab | 1.15 a | 2.42 b | 2.40 a | 4.54 a | 4.58 a | $2.12 \times 10^{-4}$ c | $3.50 \times 10^{-4}$ e |

**Table 3.** *Cont.*

| Parameters | WSC | | AC | | POM-C | | MOM-C | | Decay Rate | |
|---|---|---|---|---|---|---|---|---|---|---|
| | | | | | Soil Layer | | | | | |
| Treatments [#] | 0–15 cm | 15–30 cm | 0–15 cm | 15–30 cm | 0–15 cm | 15–30 cm | 0–15 cm | 15–30 cm | 0–15 cm | 15–30 cm |
| NG | 249.92 a | 236.54 a | 1.10 bc | 1.03 bc | 2.32 bc | 2.14 c | 3.95 c | 3.81 c | $2.42 \times 10^{-3}$ b | $7.76 \times 10^{-4}$ a |
| F | 222.70 b | 216.52 b | 0.92 d | 0.90 c | 1.95 d | 1.87 e | 3.42 d | 3.38 d | $2.31 \times 10^{-3}$ b | $1.64 \times 10^{-4}$ b |

[#] SG (50% *Festuca arundinacea* + 50% *Dactylis glomerata*), SGL1 (25% *Festuca arundinacea* + 25% *Dactylis glomerata* + 50% *Onobrychis viciifolia*), SGL2 (25% *Festuca arundinacea* + 25% *Dactylis glomerata* + 50% *Trifolium pratense*), SGL12 (25% *Festuca arundinacea* + 25% *Dactylis glomerata* + 25% *Onobrychis viciifolia* + 25% *Trifolium pratense*), and NG (natural pasture) and F (uncultivated fallow).

*3.4. Soil Carbon Mineralization Kinetics*

At the SL, C mineralization increased sharply for all land use systems after the 10th day of incubation (DAI), except SGL2. However, C mineralization increased at an increasing rate until the 38th DAI and, thereafter, increased at a steady rate for SG, SGL1, and SG12 (Figure 2). The C mineralization increased at an increasing rate until the 24th DAI and, thereafter, increased at a steady rate. At the SSL, C mineralization increased sharply for all land use systems after the 10th DAI. Although the fallow land had the lowest Ct, the proportion of C mineralization to total SOC was higher in fallow land than in others. The Ct declined by ~8, 16, and 24% under SGL1, SGL2, and SG12 over SG at the SL. The Ct under SG, SGL1, SGL2, and SG12 accounted for ~9, 7, 6, and 6% of their corresponding TOC (Figure 2). The kc declined significantly, with legume diversification indicating greater stability of SOC under SGL1, SGL2, and SG12. At the SSL, similar trends persisted for Ct and kc (Table 3).

*3.5. Enzyme Activities and Indices*

The activity of FDA under SGL1, SGL2, and SG12 increased by ~1.69 to 3.1 times over SG. Similarly, the activities of DHA, ALP, PhOX, and PerOX increased by ~1.3 to 1.5, 1.3 to 1.6, 1.5 to 2.0, and 1.4 to 1.9 times in soils under SGL1, SGL2, and SG12 over SG across the soil layers. Interestingly, the URE activity did not change significantly with legume diversification at any soil layer (Figure 3). There were significant differences among the land use systems for H and SYI indices. The highest and lowest values of H and SYI were associated with SG12 and F, respectively. Increasing legume diversification consistently enhanced the soil functional biodiversity at both layers of soil. Interestingly, SG had greater values of H and SYI than SGL1 and SGL2 but lower values than SGL12 at both soil layers (Table 4).

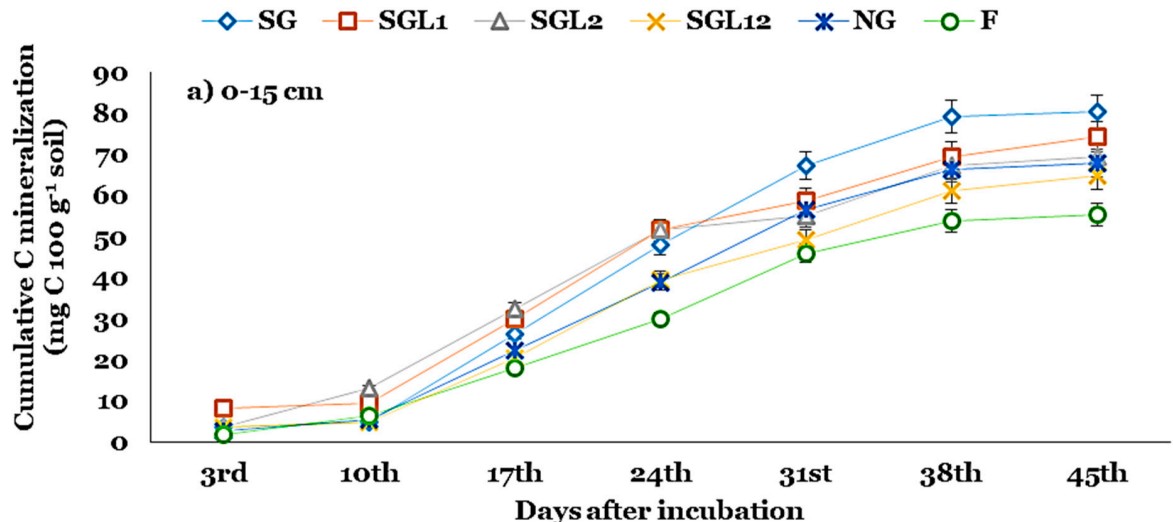

**Figure 2.** *Cont.*

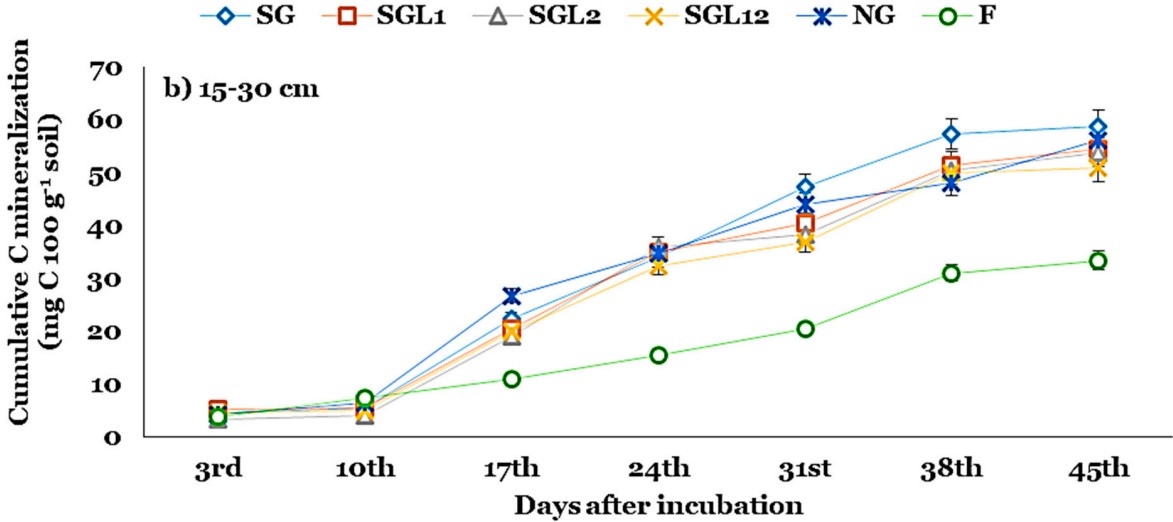

**Figure 2.** Cumulative C mineralization at the (**a**) 0–15 and (**b**) 15–30 cm soil layers in Indian Himalayan fertilized and natural pastures. SG (50% *Festuca arundinacea* + 50% *Dactylis glomerata*), SGL1 (25% *Festuca arundinacea* + 25% *Dactylis glomerata* + 50% *Onobrychis viciifolia*), SGL2 (25% *Festuca arundinacea* + 25% *Dactylis glomerata* + 50% *Trifolium pratense*), SGL12 (25% *Festuca arundinacea* + 25% *Dactylis glomerata* + 25% *Onobrychis viciifolia* + 25% *Trifolium pratense*), and NG (natural pasture) and F (uncultivated fallow).

**Table 4.** Shanon index (H), Simpson–Yule (SYI) diversity index, carbon management index (CMI), biological activity index (BAI), and ecorestoration efficiency (ERE) in Indian Himalayan fertilized and natural pastures. Values with same small-case letters are statistically similar.

| Parameters | H | | SYI | | CMI | | BAI | | ERE | |
|---|---|---|---|---|---|---|---|---|---|---|
| | | | | | **Soil Layers** | | | | | |
| Treatments [#] | 0–15 cm | 15–30 cm | 0–15 cm | 15–30 cm | 0–15 cm | 15–30 cm | 0–15 cm | 15–30 cm | 0–15 cm | 15–30 cm |
| SG | 4.42 b | 4.59 b | 4.63 c | 4.95 c | 110.40 bc | 107.52 bc | 1.38 d | 1.66 c | 1.52 c | 1.79 d |
| SGL1 | 3.91 c | 3.96 c | 4.88 bc | 5.38 b | 118.58 b | 117.56 ab | 1.94 b | 2.41 a | 2.30 b | 2.84 b |
| SGL2 | 3.76 c | 3.72 c | 5.08 b | 5.34 b | 126.93 a | 104.02 bc | 2.13 a | 2.54 a | 2.70 a | 2.64 b |
| SGL12 | 5.17 a | 5.22 a | 6.65 a | 6.21 a | 120.42 ab | 123.19 a | 2.05 ab | 2.60 a | 2.47 b | 3.20 a |
| NG | 4.12 bc | 4.18 bc | 5.03 b | 5.54 b | 115.83 b | 112.02 b | 1.65 c | 2.02 b | 1.91 c | 2.26 c |
| F | 2.83 d | 2.67 d | 1.67 d | 1.68 d | 100.00 d | 100.00 c | 1.00 e | 1.00 d | 1.00 d | 1.00 e |

[#] SG (50% *Festuca arundinacea* + 50% *Dactylis glomerata*), SGL1 (25% *Festuca arundinacea* + 25% *Dactylis glomerata* + 50% *Onobrychis viciifolia*), SGL2 (25% *Festuca arundinacea* + 25% *Dactylis glomerata* + 50% *Trifolium pratense*), SGL12 (25% *Festuca arundinacea* + 25% *Dactylis glomerata* + 25% *Onobrychis viciifolia* + 25% *Trifolium pratense*), and NG (natural pasture) and F (uncultivated fallow).

### 3.6. Stoichiometry of Soil Enzymes

The stoichiometry of enzyme activity was significantly impacted by the grass–legume mixture, indicating their significant control over soil organic matter recycling and stabilization. The enzyme stoichiometry was expressed in this study with reference to DHA activity, such as DHA: FDA, DHA: URE, DHA: ALP, DHA: PhOX, and DHA: PerOX. For example, the DHA: FDA value of 1.47 indicates that for 1.47 units of DHA activity, 1 unit of FDA activity would happen, implying that the narrower the ratio, the lesser the decomposition of SOM and the better the stability of C in soil. In this study, legume inclusion significantly narrowed the ratios for DHA: FDA, DHA: ALP, DHA: PhOX, and DHA: PerOX but had no impact on DHA: URE at both layers of soil. Interestingly, the SG had wider values for these ratios than NG, indicating poor C stabilization under SG (Figure 4).

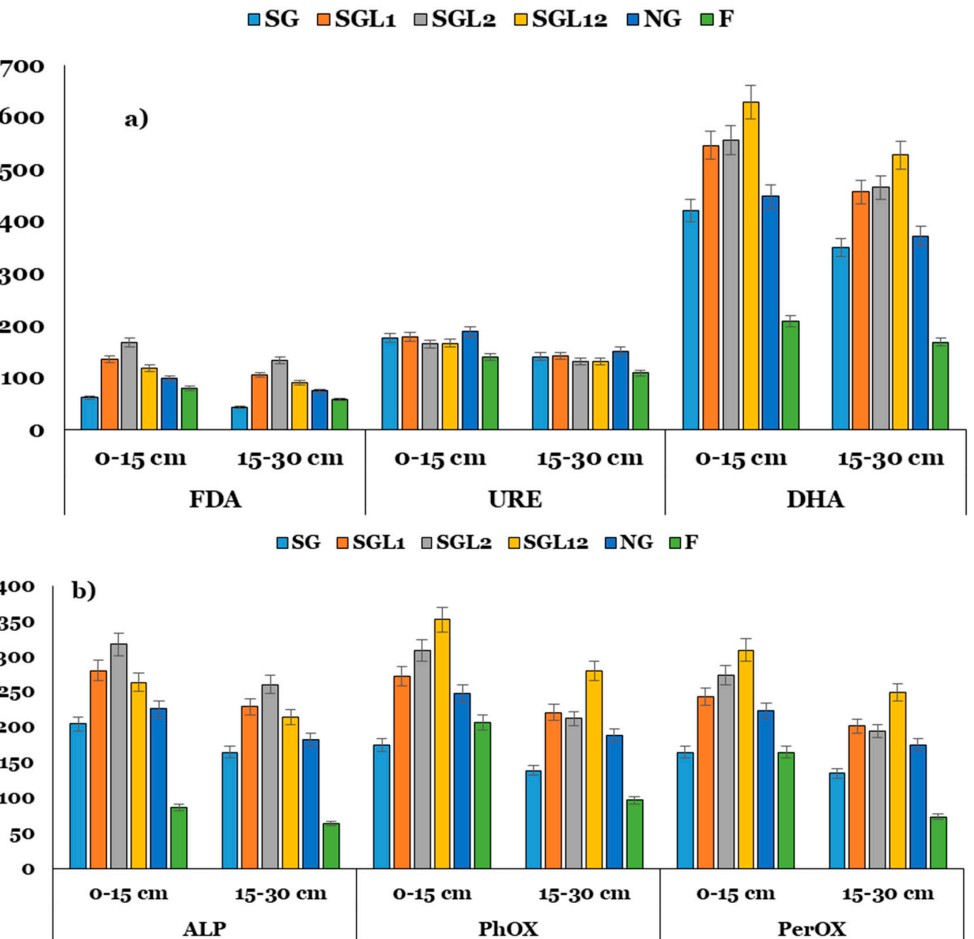

**Figure 3.** Activities of FDA (µg fluorescein $g^{-1}$ $h^{-1}$), urease (URE; µg $NH_4^+$ $g^{-1}$ $h^{-1}$), dehydrogenase (DHA; µg TPF $g^{-1}$ 24 $h^{-1}$) and alkaline phosphatase (ALP; µg PNP $g^{-1}$ $h^{-1}$), phenol oxidase (PhOX; µg ABTS $g^{-1}$ $h^{-1}$), peroxidase (PerOX; µg ABTS $g^{-1}$ $h^{-1}$) at the (**a**) 0–15 and (**b**) 15–30 cm soil layers in Indian Himalayan fertilized and natural pastures. The LSD is denoted by error bars. SG (50% *Festuca arundinacea* + 50% *Dactylis glomerata*), SGL1 (25% *Festuca arundinacea* + 25% *Dactylis glomerata* + 50% *Onobrychis viciifolia*), SGL2 (25% *Festuca arundinacea* + 25% *Dactylis glomerata* + 50% *Trifolium pratense*), SGL12 (25% *Festuca arundinacea* + 25% *Dactylis glomerata* + 25% *Onobrychis viciifolia* + 25% *Trifolium pratense*), and NG (natural pasture) and F (uncultivated fallow).

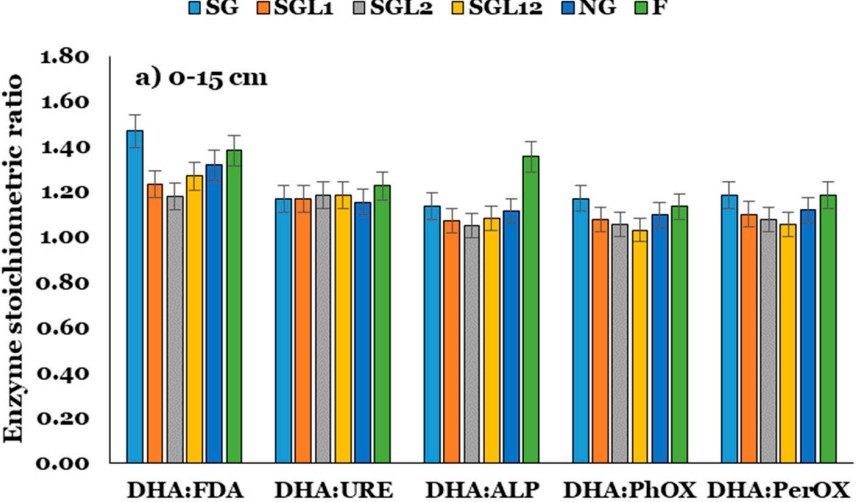

**Figure 4.** *Cont.*

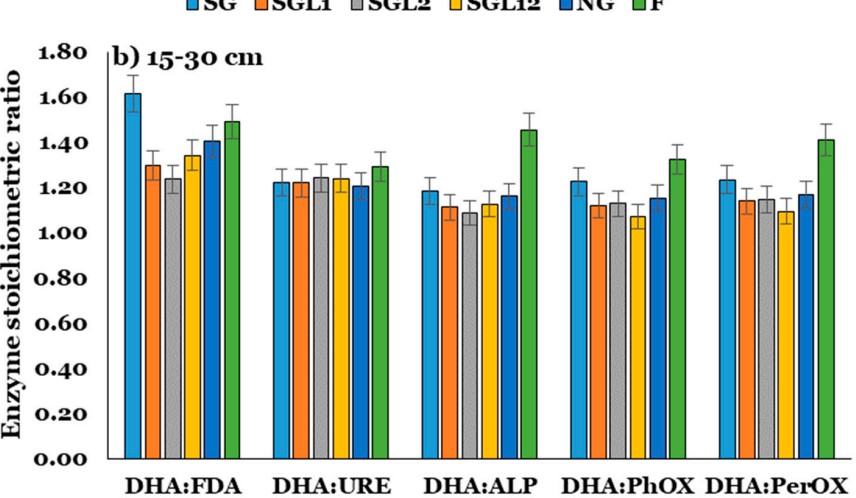

**Figure 4.** Stoichiometric ratio of enzymes at the (**a**) 0–15 and (**b**) 15–30 cm soil layers in Indian Himalayan fertilized and natural pastures. The LSD is denoted by error bars. SG (50% *Festuca arundinacea* + 50% *Dactylis glomerata*), SGL1 (25% *Festuca arundinacea* + 25% *Dactylis glomerata* + 50% *Onobrychis viciifolia*), SGL2 (25% *Festuca arundinacea* + 25% *Dactylis glomerata* + 50% *Trifolium pratense*), SGL12 (25% *Festuca arundinacea* + 25% *Dactylis glomerata* + 25% *Onobrychis viciifolia* + 25% *Trifolium pratense*), and NG (natural pasture) and F (uncultivated fallow).

### 3.7. Soil Organic Carbon Accumulation and Sequestration

The accumulation and sequestration of SOC were significantly impacted by grass–legume diversification and fertilizer application. The C accumulation at 0–30 cm soil layer under SGL1, SGL2, and SG12 was ~19, 12, and 26% higher than SG, respectively. The C accumulation under NG was lower by ~10% SG, which might also be due to fertilizer application. However, C sequestration under SGL1, SGL2, and SG12 was ~27, 22, and 38% higher than SG, respectively, at the 0–30 cm soil layer (Table 5). The C sequestration under NG was lower by ~21% SG. The C accumulation and sequestration under fallow was significantly lower than other land use systems. Approximately 45% of SOC was sequestered across the land use systems.

**Table 5.** Carbon accumulation and sequestration (Mg C ha$^{-1}$) at 0–30 cm soil layer in Indian Himalayan fertilized and natural pastures. Values with same small-case letters are statistically similar.

| Treatments [#] | C Accumulation (Mg C ha$^{-1}$) | C Sequestration (Mg C ha$^{-1}$) |
| --- | --- | --- |
| SG | 34.97 c | 14.72 c |
| SGL1 | 41.48 ab | 18.66 b |
| SGL2 | 39.33 bc | 17.93 b |
| SGL12 | 44.03 a | 20.31 a |
| NG | 38.57 bc | 17.88 b |
| F | 27.94 d | 12.77 d |

[#] SG (50% *Festuca arundinacea* + 50% *Dactylis glomerata*), SGL1 (25% *Festuca arundinacea* + 25% *Dactylis glomerata* + 50% *Onobrychis viciifolia*), SGL2 (25% *Festuca arundinacea* + 25% *Dactylis glomerata* + 50% *Trifolium pratense*), SGL12 (25% *Festuca arundinacea* + 25% *Dactylis glomerata* + 25% *Onobrychis viciifolia* + 25% *Trifolium pratense*), and NG (natural pasture) and F (uncultivated fallow).

### 3.8. Carbon Management Index (CMI), Biological Activity Index (BAI), and Ecorestoration Efficiency (ERE)

Legume diversification significantly improved the indicators of ecosystem functions, such as carbon management index (CMI), biological activity index (BAI), and ecorestoration efficiency (ERE). The CMI under SGL1, SGL2, and SG12 was ~9, 27, and 20% higher than SG

at the SL. They enhanced the BAI by 1.5 times over SG at both layers of soil. They enhanced the ERE by ~50, 77, and 62% over SG, respectively, at the SL. Interestingly, the values of CMI, BAI, and ERE were significantly lower under SG as compared to NG, indicating ecosystem degradation (Table 4).

## 4. Discussion

*4.1. Legume Diversification Shapes Soil Functional Diversity and Enzyme Stoichiometry*

Among the experimental plots that received the same fertilizer, legumes considerably enhanced biomass in this study. This acceleration might be explained by three unique mechanisms: (1) the capacity of legumes to absorb nitrogen, (2) the favorable interactions with non-legume species, and (3) the encouragement of community diversity. Legumes encourage C additions' greater soil carbon storage and biomass (Tables 1 and 5). Second, niche divergence and facilitation may have both contributed to the complementarity that resulted from interactions between species of legumes and non-legumes [17]. Finally, since they complement grasses and are more likely to occur in grasslands with high levels of biodiversity, legumes have an impact [18]. With changes in community functional composition brought on by diversification, the benefits of species diversity on productivity can be realized [32].

With increasing legume diversification, the functional diversity of soil increased due to substrate diversity and microbial diversity arising due to diversity in rhizodeposition. The greater activities of DHA, FDA, and ALP could be linked to the higher SOC content and greater biomass production (Figure 3). The decline in URE activity under SGL1, SGL2, and SGL12 could be due to increasing N availability from N fixation by legumes. The activity of C-degrading enzymes such as PhOX and PerOX increased with legume diversification due to the greater concentration of recalcitrant C (Table 1).

Given the differences in characteristics and unique ecological functions between grasses and legumes [33], grass–legume combinations are anticipated to show an increased niche partition. This may intensify the impacts of complementarity between various plant species, enhancing the functional diversity of the soil, plant production, and soil labile and recalcitrant C fluxes [34]. Yet, if both labile and recalcitrant resources are present, bacteria selectively choose labile substrates to reduce their energy expenditure [35]. Increased DHA and FDA activity in the soil is, thus, predicted as a result of a higher grass–legume combination. This emphasizes how crucial it is to encourage legume diversity in order to improve ecosystem functions, such as BAI, CMI, and ERE (Table 4). In a grass–legume mixture, the N availability of legumes is met by N fixation, but grasses scavenge N from organic recalcitrant molecules [36,37]; the increased decomposition of C degrading enzymes, namely, PhOX and PerOX, is associated with an increasing grass–legume mixture. The NMDS analysis also revealed that PhOX and PerOX activity is dependent on MBC, ALP, and URE (Figure 5). The greater activities of ALP under the legume mixture also facilitate substrate accessibility for PhOX and PerOX. The larger active microflora under the grass–legume combination was also suggested by the increased microbial biomass C, DHA, and FDA (Table 2 and Figure 3). In addition, soil pH and enzyme activity are closely related. Under pH 7.3, PhOX and PerOX were observed to have increased activity [38]. The pH of the soil was kept below 7.3 by diversifying the leguminous crops.

The narrower ratios of DHA: FDA, DHA: ALP, DHA: PhOX, and DHA: PerOX indicated the significant impact of legumes on microbial activity. Ratios of DHA: PhOX and DHA: PerOX significantly decreased with an increasing grass–legume mixture, suggesting lesser microbial demand for carbon [39]. Further, SOC had a positive influence on DHA, PhOX, and PerOX activity. As per the resource allocation theories, more N might reduce the activity of the enzymes that help bacteria acquire nitrogen, which would result in more bacteria collecting carbon [40]. Living organisms require C and N as substrates and nutrients in order to produce enzymes and boost output [41]. Moreover, in the existence of an excess of N, the growth of microorganisms and their functions may also be impeded [42,43]. Legume diversification decreased URE activity due to N supply from biological fixation (Figure 4).

The microbiological economic allocation hypothesis, which states that microorganisms typically devote more resources to manufacture the enzymes to capture the scarcest elements, is supported by this information [44]. As N was efficiently provided, P and C would have greater limitations than N. Microbes will, therefore, devote more resources to acquiring P and C while devoting less resources to acquiring N, leading to a greater activity of ALP, PhOX, and PerOX under SGL1, SGL2, and SGL12 (Figure 3) [45]. Clearly, based on NMDS, the legume diversification impacted the microbial enzyme activities and their respective stoichiometries (Figure 5).

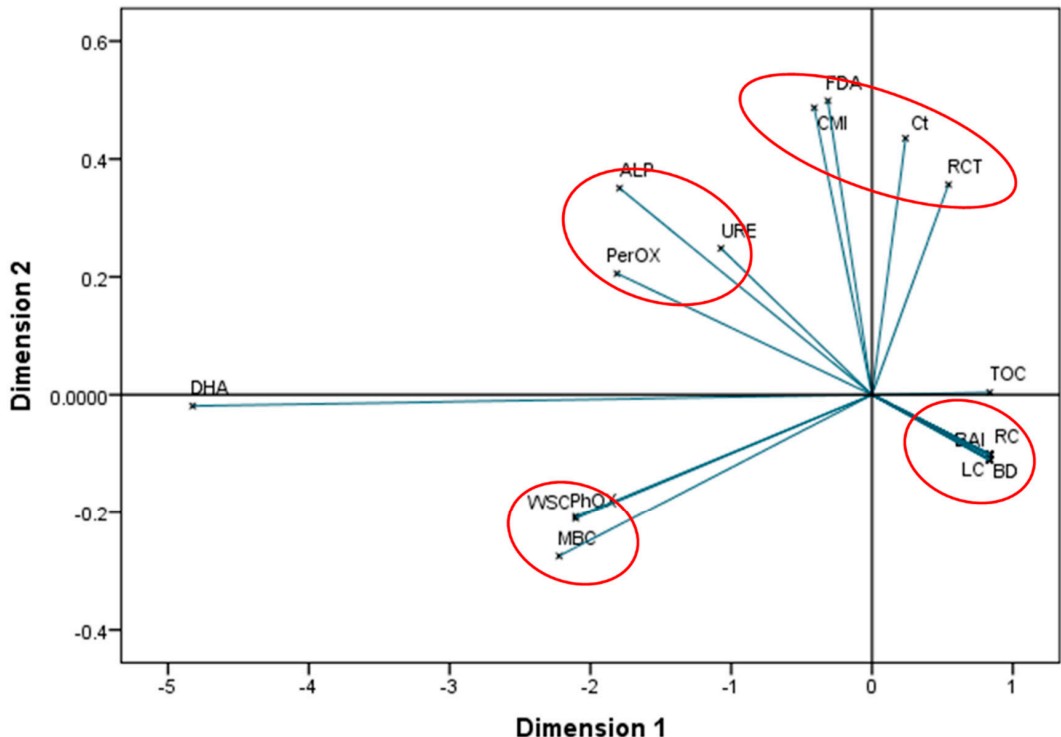

**Figure 5.** NMDS analysis of soil enzyme activity and carbon pools of different pasture combinations of Indian Himalayan fertilized and natural pastures. BD: bulk density; RC: recalcitrant C, LC: labile C; BAI: biological activity index; WSC: water-soluble C; PhOX: phenoloxidase activity; MBC: microbial biomass c; ALP: alkaline phosphatase activity; URE: urease activity; Perox: peroxidase activity; FDA: fluorescein diacetate activity; C < I: carbon management index; Ct: carbon mineralization; RCT: ratio of RC to TOC. Other abbreviations are same as described in text.

### 4.2. Legume Diversification Regulate Soil Carbon Pools and C Mineralization and Sequestration

The key reason for the greater concentration of various carbon components under SGL1, SGL2, and SGL12 was the greater carbon intake since around 43–47% of the legume roots remained inert and added SOC after the harvest. Labile C fractions are much more responsive to alter with legume diversity and fertilizer application, as seen by the enhanced AC from legume incorporation over SG and NG. It is pretty intriguing that the study found a discernible boost in recalcitrant C under SGL1, SGL2, and SGL12 over SG [46]. This suggested that legume was critical to enhance the soil resistant C, which persists longer in soil, in addition to being essential to raise the SOC stock. Data from the CMI also showed that a feasible solution to increase the C in the soil is legume diversification in temperate pasture systems. The major carbon reservoir in the soil under research was the labile carbon pool. This may be due to the ecology of grasslands and the properties of the soil.

The incorporation of grass and legume residues improved soil carbon fractions under SGL1, SGL2, and SGL12. Higher CMI was directly linked to higher carbon fractions under legume diversification (Table 4). The addition of legumes to cereal-based farming enhanced the CMI, according to [47]. Leguminous residues with a low carbon/nitrogen ratio may

be more beneficial for boosting C stabilization in temperate soils, according to the study's increased C sequestration under SGL12. Decreasing carbon supply was the major driver of the drop in C sequestration in SG and NG (Figure 2). Despite greater C mineralization in legume-diversified plots, the proportion of C mineralization to total SOC under these plots was significantly lower than those of F, resulting in a greater accumulation. Moreover, greater biomass production in legume-diversified systems might have enhanced the C supply and, subsequently, C sequestration [16,29,46]. Under SGL1, SGL2, and SGL12, the variety of C sources supplied by legumes and grasses caused a rise in the functional biodiversity of the soils (H and SYI). According to the PCA analysis, C sequestration was directly influenced by C pools, enzyme activity, and microbial activity, but under legume diversification, enzymatic stoichiometry had an indirect but substantial impact on C sequestration (Figure 6). The correlation analysis found that enzyme stoichiometry significantly affects C accumulation, sequestration, and mineralization. It showed a substantial negative association between DHA: PerOX, DHA: PhOX, and C sequestration as a result of legume diversification (Table 6). The basic process of carbon sequestration during legume evolution was made clear by this. The lower ratios of DHA: PerOX and DHA: PhOX revealed reduced activity of PerOX and PhOX relative to DHA per unit of substrate or RC availability, suggesting increased RC and MOMC accumulation and C sequestration (Figure 4; Tables 1 and 2). Similar to this, reduced RC decomposition under legume diversification in temperate pasture was shown by the substantial negative connection between DHA: PerOX, DHA: PhOX, and C mineralization.

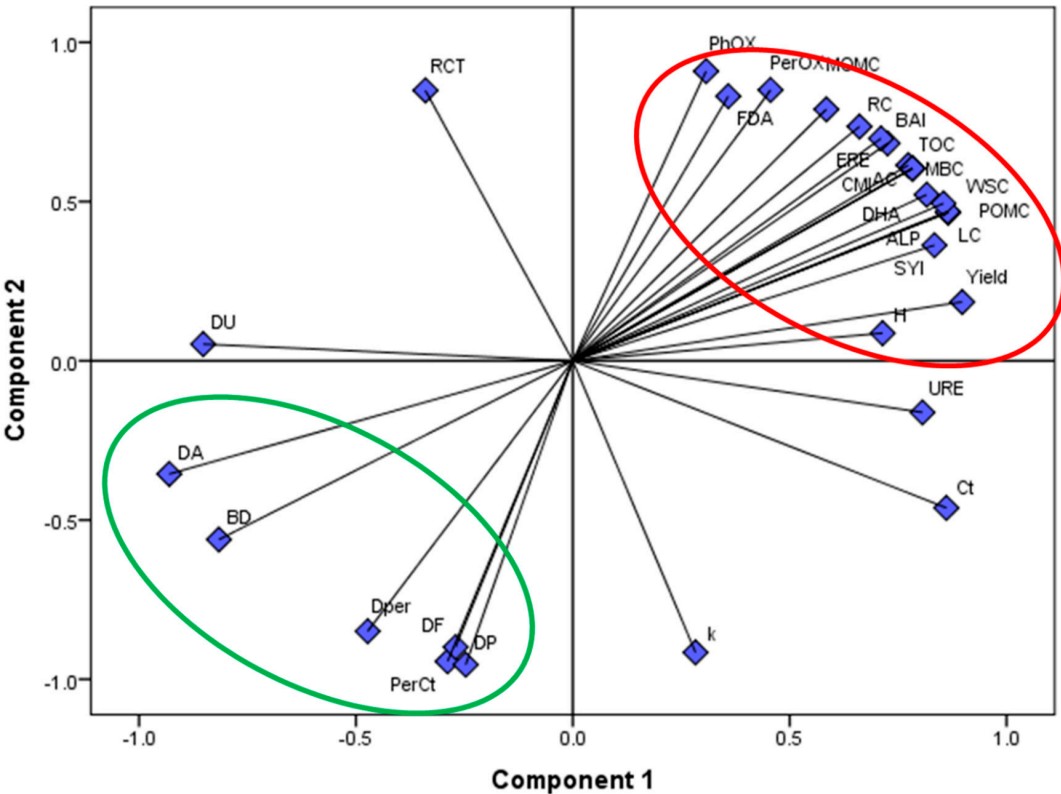

**Figure 6.** PCA indicating the impact of legume diversification on soil carbon sequestration as mediated by C pools and enzyme stoichiometric ratios in Indian Himalayan fertilized and natural pastures. BD: bulk density; RCT: ratio of RC to TOC; DA: DHA: ALP; Dper: DHA: PerOX; DF: DHA:FDA; DP: DHA: PhOX; PerCt: percent C mineralization with respect to TOC. BD: bulk density; RC: recalcitrant C, LC: labile C; BAI: biological activity index; WSC: water-soluble C; PhOX: phenoloxidase activity; MBC: microbial biomass C; ALP: alkaline phosphatase activity; URE: urease activity; Perox: peroxidase activity; FDA: fluorescein diacetate activity; C < I: carbon management index; Ct: carbon mineralization; RCT: ratio of RC to TOC.

**Table 6.** Correlation analysis indicating the relationship among enzyme stoichiometric ratios and C accumulation, sequestration, and mineralization in Indian Himalayan fertilized and natural pastures (NS: non-significant; * significant at $p < 0.05$; **: significant at $p < 0.01$).

| Variables | DHA:FDA | DHA:URE | DHA:ALP | DHA: PhOX | DHA: PerOX |
|---|---|---|---|---|---|
| C accumulation | −0.592 [NS] | −0.642 * | −0.771 * | −0.796 ** | −0.801 ** |
| C sequestration | −0.634 * | −0.517 [NS] | −0.738 * | −0.799 ** | −0.792 ** |
| C mineralization | 0.287 [NS] | 0.935 ** | 0.678 * | 0.559 [NS] | 0.627 * |

DHA: dehydrogenase activity; PhOX: phenoloxidase activity; ALP: alkaline phosphatase activity; URE: urease activity; PerOX: peroxidase activity; FDA: fluorescein diacetate activity.

Following harvest, the root remains in the soil and acts as a carbon supplier. By absorbing atmospheric N2 through symbiotic relationships with rhizobia, legumes, unlike other crops, may improve plant biomass and N content without absorbing excessive amounts of soil nutrients [48]. In addition to improving ecosystem services, biological N fixation also has a significant impact on SOC dynamics. In the present study, SGL1, SGL2, and SGL12 increased the MBC, AC, LC, and RC concentrations, indicating that legume diversification could be an effective measure for increasing SOC levels (Tables 1 and 2; [48]).

The main energy supply for soil microbial activity is provided by essential labile organic carbon components (AC and POMC), and MBC is a key indication for describing microbial load and activity [43]. Due to a change in the stoichiometric ratio of soil enzymes, legume diversity raised the MBC and AC contents of the soil in the current research [31]. The current study demonstrated that legume diversity greatly boosted the contents of LC and AC, which were primarily composed of fresh and plant debris. Legume production has positive impacts on soil nutrient accessibility and microbiological community structure, which can boost the production of other crops in the agricultural system and, as a result, improve the amount of carbon supply to the agroecosystem. Therefore, the growth of labile organic carbon fractions generated from plants is facilitated by the growing of legumes. Ultimately, legume inclusion modifies the soil microbiome's structural makeup to promote C sequestration in soil.

## 5. Conclusions

Our results revealed that legume diversity has a significant impact on soil C storages in temperate grasslands. The above-ground biomass, vegetation composition, microbiological biomass, soil functional diversity, and soil enzyme stoichiometry were the main variables linked to larger soil C storages. Legumes improved biomass output, which led to a rise in soil carbon stocks. Legumes also changed the stoichiometric ratio of soil enzymes and enhanced soil labile and recalcitrant C at the same time. In the meantime, C sequestration in temperate grasslands was considerably aided by the stoichiometric correlations of enzyme activity. The presence of legumes dramatically changed soil C storage by altering the soil enzyme stoichiometric ratio. Soils under SGL1, SGL2, and SG12 had ~18, 36, and 22% greater soil C than SG, respectively. The C mineralization was suppressed by legume diversification. C sequestration under SGL1, SGL2, and SG12 was ~27, 22, and 38% higher than SG, respectively, at the 0–30 cm soil layer. Under legume diversification, enzymatic stoichiometry had an indirect but substantial impact on C sequestration (Figure 6). The correlation analysis found that enzyme stoichiometry significantly affects C accumulation, sequestration, and mineralization. These findings shed light on the factors that regulate the cycling of nutrients and soil C in temperate grassland ecosystems. In the temperate zones, it is proposed that the coexistence of grass and legume species may significantly improve ecosystem functions such as soil C storage, productivity, and diversity.

**Author Contributions:** Conceptualization, A.G. and S.A.; methodology, A.G. and S.A.; validation, A.K.S., P.J. and R.K.Y.; formal analysis, A.G.; investigation, S.A., S.S.B. and N.H.M.; data curation, A.G.; writing—original draft preparation, A.G.; writing—review and editing, R.J.B. and N.B. All authors have read and agreed to the published version of the manuscript.

**Funding:** This research received no external funding.

**Data Availability Statement:** Data are contained within the article.

**Acknowledgments:** The authors express sincere gratitude to the competent authority of ICAR-IGFRI for facilitating the work.

**Conflicts of Interest:** The authors declare no conflicts of interest.

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
