# Peer review of "Soil Carbon Storage, Enzymatic Stoichiometry, and Ecosystem Functions in Indian Himalayan Legume-Diversified Pastures"

_land, doi:10.3390/land13040452_

Round 1

Reviewer 1 Report

Comments and Suggestions for Authors

This study is interesting. However, a few major constraints need to be rectified.

Introduction: What are the scientific hypotheses of this study?

Materials and Methods: What tools were used to collect the soil samples? Methods for determination of soil samples needs more clarity. What is the method of correlation analysis? The data analysis needs further clarification.

Results : I suggest a three-line table for this paper.

Discussion: The discussion is poor, and it needs to be described in more detail.

Author Response

This study is interesting. However, a few major constraints need to be rectified. Introduction: What are the scientific hypotheses of this study? Response: We hypothesized that increasing legume variety would change the microbial activity and enzyme stoichiometry. Materials and Methods: What tools were used to collect the soil samples? Methods for determination of soil samples needs more clarity. What is the method of correlation analysis? The data analysis needs further clarification. Response: We have now mentioned clearly. Soil samples were taken from surface layer (SL: 0-15 cm) and sub surface layer (SSL: 15-30 cm) during March, 2022 using a core sampler. Pearson Correlation was analyzed among enzyme stoichiometries with C mineralization (Ct), accumulation (Cacc), sequestration (Cseq). The data were subjected to analysis of variance as applicable to RBD using Tukey’s test for mean differentiation. The following model was used yij = µ+ τi + bj + ϵij, where μ is the general mean; τi are the main effects of treatments, bj are the effects of blocks, and ϵij is a usual independent and identically distributed error term with expectation zero and variance σ2. Results : I suggest a three-line table for this paper. Response: The tables have been reformatted. Discussion: The discussion is poor, and it needs to be described in more detail. Response: We have substantially revised the discussion.

Reviewer 2 Report

Comments and Suggestions for Authors

The paper is interesting and brings up once again the favourable effect legumes have on soil and carbon storage. There is a lot of literature related to this topic, my suggestion is to use it more effectively, especially in the Discussion part. This section should compare previous research with your own results, and in this paper there are more statements than comparative discussions.

In the Material and Methods part please specify the exact period of experimentation, between which years (2016-2022? the period is not clear). The climate description must be for the experimental period. The description of the experiment needs to be made clearer, it is quite difficult to identify which are the experimental variants, sizes, treatments, etc. A schematic presentation of the experimental design would be useful.

In the Results part please explain all abbreviations in tables and figures by footnotes. 

In addition, please pay attention to editing, layout, typos. Please respect the requirements of the journal.

Author Response

The paper is interesting and brings up once again the favourable effect legumes have on soil and carbon storage. There is a lot of literature related to this topic, my suggestion is to use it more effectively, especially in the Discussion part. This section should compare previous research with your own results, and in this paper there are more statements than comparative discussions.

In the Material and Methods part please specify the exact period of experimentation, between which years (2016-2022? the period is not clear). The climate description must be for the experimental period. The description of the experiment needs to be made clearer, it is quite difficult to identify which are the experimental variants, sizes, treatments, etc. A schematic presentation of the experimental design would be useful.

Response: The long-term experiment was initiated in 2016 and soil sample for the impact analysis was collected in 2022. Climate has been described for the experimental period. Plot size was 8m*8m. A schematic presentation of the experimental design has been attached in Fig 1. We have substantially revised the discussion.

In the Results part please explain all abbreviations in tables and figures by footnotes. 

Response: Added

In addition, please pay attention to editing, layout, typos. Please respect the requirements of the journal.

Response: We have substantially revised the editing using an AI software.

Reviewer 3 Report

Comments and Suggestions for Authors

The work is devoted to the key topic of the influence of species diversity on ecosystem functions. It has the potential to contribute to the already extensive body of research on this issue. However, as presented, the work suffers from a fundamental methodological flaws that must be corrected for publication.

It is incorrect to establish in an experiment the influence of one factor (the number of legume species) while another factor (fertilizers) is changing. If you had in mind some more complex experimental approach, when two factors change at once, you need to clearly describe it. In the presented results there is also no clear difference between the influence of these two factors.

In their experiment, the authors consider a set of six types of plant communities, four of which received fertilizers, and the other two (NG and F) did not. In order to compare indicators of ecosystem functions to identify the influence on them certain species, communities must be in the same conditions. The authors violated this principle of experiment. The identified differences in ecosystem functions can be explained by both species composition and fertilization.

There are two ways to solve this problem:

1) remove from the article all data on pasture types NG and F, and compare only those four types that received the same fertilizer

OR

2) somehow take into account the effect of fertilizer on the ecosystem functions studied by the authors (analysis of additional data, correlations, literary information, etc.)

For minor but not less important shortcomings related to the description of methods and results, see detailed comments below

Detailed comments. See also uploaded PDF file

Line 23 "Five types" or "five species combinations"

Line 29 Section 3.4 does not show this. The lowest mineralization was revealed in pasture type F in which the presence of legumes is not explicitly indicated by the authors.

Line 44 Reference [1] is an old (2007) FAO report on water. Please check the link

Lines 79-80 References [10-12] are, of course, classic works on the effects of species diversity on ecosystem functions, but they are already 15 to 20 years old. Since then, more recent reviews have appeared on this topic.

Line 116 It's better to start a new paragraph here

Line 124 What is the species composition of P and S? Are there legumes in there? What do the first four options compare to?

Lines 124-129 Do you mean the first four combinations here? Please clarify

Line 129 Identical to what?

Lines 132-135 If this is the approximate species composition of NG, then what is it for F? And it is necessary to explicitly indicate whether there were legumes there or not

Lines 136-137 It is incorrect to establish in an experiment the influence of one factor (the number of legume species) while another factor (fertilizers) is changing. If you had in mind some more complex experimental approach, when two factors change at once, you need to clearly describe it. In the presented results there is also no clear difference between the influence of these two factors.

Lines 224-239 Section 3.4. Nothing is written about the values for NG and F, although from Fig. 2 it can be seen that F is characterized by minimal C mineralization, which is important for the stated research tasks

Line 226. Figure 2 doesn't show this

Lines 265-269 This applies more to Methods

Line 292 It is better to change this subtitle, since previous results were also about ecosystem functions

Line 306 There are no correlation coefficients in these tables. You have not identified a correlation between these indicators in presented text

Lines 308-311 These studies [17, 18, 32] are 15 to 20 years old. No more recent research?

Lines 359-360 To compare this with NG and F you must first prove that the increase in C intake is not a result of the application of fertilizers

Lines 375-376 This needs to be explained, since Fig. 2 shows С mineralization. If it is considered a factor in decreasing C supply, then explanations are still needed, since the indicators of C mineralization for NG are among those for the experimental combinations with legumes, and the mineralization in F is the lowest

Comments on the Quality of English Language

English is fine

Author Response

The work is devoted to the key topic of the influence of species diversity on ecosystem functions. It has the potential to contribute to the already extensive body of research on this issue. However, as presented, the work suffers from a fundamental methodological flaws that must be corrected for publication.

Response:  Thank you very much for your comment.

It is incorrect to establish in an experiment the influence of one factor (the number of legume species) while another factor (fertilizers) is changing. If you had in mind some more complex experimental approach, when two factors change at once, you need to clearly describe it. In the presented results there is also no clear difference between the influence of these two factors.

Response: Thank you very much for your comment. In the study region, local farmers commonly use natural pasture as source of fodder to animals. We aimed to understand the impact of legume diversification with recommended input supply, including irrigation, fertilizer. Hence, we have compared the results with natural pasture (without fertilizer application) and fallow (without crop).  Now we have specified this in the manuscript also.

In their experiment, the authors consider a set of six types of plant communities, four of which received fertilizers, and the other two (NG and F) did not. In order to compare indicators of ecosystem functions to identify the influence on them certain species, communities must be in the same conditions. The authors violated this principle of experiment. The identified differences in ecosystem functions can be explained by both species composition and fertilization.

There are two ways to solve this problem:

1) remove from the article all data on pasture types NG and F, and compare only those four types that received the same fertilizer

OR

2) somehow take into account the effect of fertilizer on the ecosystem functions studied by the authors (analysis of additional data, correlations, literary information, etc.)

Response: In the study region, local farmers commonly use natural pasture as source of fodder to animals. We aimed to understand the impact of legume diversification with recommended input supply, including irrigation, fertilizer. Hence, we have compared the results with natural pasture (without fertilizer application) and fallow (without crop). Now we have specified this in the manuscript also.

For minor but not less important shortcomings related to the description of methods and results, see detailed comments below

 Detailed comments. See also uploaded PDF file

Line 23 "Five types" or "five species combinations"

Response: Thank you very much for your comment. We change by “Five types”

Line 29 Section 3.4 does not show this. The lowest mineralization was revealed in pasture type F in which the presence of legumes is not explicitly indicated by the authors.

Response: Thank you very much for your comment. We have written as “ Among the pastures, the C mineralization was suppressed by legume diversification.”

Line 44 Reference [1] is an old (2007) FAO report on water. Please check the link

Response: Thank you very much for your comment. Sorry, we have rectified now.

Lines 79-80 References [10-12] are, of course, classic works on the effects of species diversity on ecosystem functions, but they are already 15 to 20 years old. Since then, more recent reviews have appeared on this topic.

Response: Thank you very much for your comment. We have updated with recent references.

Line 116 It's better to start a new paragraph here

Response: Thank you very much for your comment. Done.

Line 124 What is the species composition of P and S? Are there legumes in there? What do the first four options compare to?

Response: Thank you very much for your comment. The natural pastures comprised of Bromus spp., Phalaris spp., Brachiaria spp., Paspalam spp., Chrysopogon spp., Bothriocloa spp., Setaria spp. The uncultivated fallow does not contain any crop and grass.

Lines 124-129 Do you mean the first four combinations here? Please clarify

Response: Thank you very much for your comment. Yes.

Line 129 Identical to what?

Response: Thank you very much for your comment. Identical in terms of slope, topography, and with similar soil texture and parent material

Lines 132-135 If this is the approximate species composition of NG, then what is it for F? And it is necessary to explicitly indicate whether there were legumes there or not

Response:  Thank you very much for your comment. The natural pastures comprised of Festuca arundinacea, Dactylis glomerata, Bromus spp., Phalaris spp., Brachiaria spp., Paspalam spp., Chrysopogon spp., Bothriocloa spp., Setaria spp., Chenopodium glaucum, Capsella bursa-pastoris, Plantogo lanceolata, Daucus carota, Convolvulus arvensis etc low yielding grasses (Poa spp., Agropyron spp., Bromus spp., Alopecurus spp., Avena fatua, etc.) and shrubs like Rubus spp. and Zizyphus spp. The uncultivated fallow does not contain any crop and grass.

Lines 136-137 It is incorrect to establish in an experiment the influence of one factor (the number of legume species) while another factor (fertilizers) is changing. If you had in mind some more complex experimental approach, when two factors change at once, you need to clearly describe it. In the presented results there is also no clear difference between the influence of these two factors.

Response: Thank you very much for your comment. In the study region, local farmers commonly use natural pasture as source of fodder to animals. We aimed to understand the impact of legume diversification with recommended input supply, including irrigation, fertilizer. Hence, we have compared the results with natural pasture (without fertilizer application) and fallow (without crop). Now we have specified this in the manuscript also.

Lines 224-239 Section 3.4. Nothing is written about the values for NG and F, although from Fig. 2 it can be seen that F is characterized by minimal C mineralization, which is important for the stated research tasks

Response: Thank you very much for your comment. Although the fallow land had the lowest Ct, proportion of C mineralization to total SOC was higher under fallow than others.

Line 226. Figure 2 doesn't show this

Response: Thank you very much for your comment. Figure 2 indicated Ct and Table 3 indicated Kc.

Lines 265-269 This applies more to Methods

Response: Thank you very much for your comment. We moved it to the Methods section.

Line 292 It is better to change this subtitle, since previous results were also about ecosystem functions

Response: Thank you very much for your comment. Done!

Line 306 There are no correlation coefficients in these tables. You have not identified a correlation between these indicators in presented text

Response: Thank you very much for your comment. We deleted correlation.

Lines 308-311 These studies [17, 18, 32] are 15 to 20 years old. No more recent research?

Response: Thank you very much for your comment. We are inclined to mention these works because they are the pioneering basis of this knowledge.

Lines 359-360 To compare this with NG and F you must first prove that the increase in C intake is not a result of the application of fertilizers

Response: Thank you very much for your comment. Of course, they are not because of fertilization as fertilizer was added to fallow but SOC did not increase.

Lines 375-376 This needs to be explained, since Fig. 2 shows Ð¡ mineralization. If it is considered a factor in decreasing C supply, then explanations are still needed, since the indicators of C mineralization for NG are among those for the experimental combinations with legumes, and the mineralization in F is the lowest

Response: Thank you very much for your comment. Despite, greater C mineralization in legume diversified plots, the proportion of C mineralization to total SOC under these plots were significantly lower than those of F, resulting a greater accumulation. Moreover, greater biomass production in legume diversified systems might have enhanced the C supply and subsequently C sequestration

Reviewer 4 Report

Comments and Suggestions for Authors

The purpose of this study was to investigate the influences of legume diversification on activities and stoichiometry of soil enzymes and their relation to C sequestration in Himalayan pasturelands. The following are noted:

line 113: pH 6.7- is near to neutral

line 114-115, “low in total organic carbon (6.9 g kg-1), nitrogen (280 kg ha-1), phosphorus (17 kg 114 ha-1), and high in potassium (468 kg ha-1). Why are the characteristics of  indicators differently presented for the organic carbon g kg-1, for other indicators kg ha-1?

lines 149-152 Describe in more detail the analytic methods used, not only by pointing to references:

 “Total soil organic carbon (TOC), labile and recalcitrant carbon (LC and RC), [23], particulate and mineral organic matter associated C (POMC and MOMC), [24], water soluble C (WSC) [25], and active C (AC) [26] were estimated for each soil samples. Carbon mineralization from the soil samples were measured for 45 days using the incubation, followed by entrapment method [16].”

line 153 decay “kiunetics”, kinetics??? Mistake

line 253 μg fluorescein??? Mistake

Line 260 mnagmenet??? Mistake

 Table 6. An empty column in the table?....In addition, questions arise about the meaning of correlations in this table, since they are all negative, although the processes of mineralization and sequestration are of opposite directions in their essence.

In the Conclusions section, I would suggest specifying in more details the influence of legumes and specific grasses combinations on carbon storage, enzymatic stoichiometry, etc.

Comments on the Quality of English Language

I recommend English editing.

Author Response

The purpose of this study was to investigate the influences of legume diversification on activities and stoichiometry of soil enzymes and their relation to C sequestration in Himalayan pasturelands. The following are noted:

 line 113: pH 6.7- is near to neutral

Response: We have added now.

line 114-115, “low in total organic carbon (6.9 g kg-1), nitrogen (280 kg ha-1), phosphorus (17 kg 114 ha-1), and high in potassium (468 kg ha-1). Why are the characteristics of  indicators differently presented for the organic carbon g kg-1, for other indicators kg ha-1?

Response: We have converted to kg per ha

lines 149-152 Describe in more detail the analytic methods used, not only by pointing to references:

 “Total soil organic carbon (TOC), labile and recalcitrant carbon (LC and RC), [23], particulate and mineral organic matter associated C (POMC and MOMC), [24], water soluble C (WSC) [25], and active C (AC) [26] were estimated for each soil samples. Carbon mineralization from the soil samples were measured for 45 days using the incubation, followed by entrapment method [16].”

Response: We have described detail the analytic methods used.

line 153 decay “kiunetics”, kinetics??? Mistake

Response: Corrected.

line 253 μg fluorescein??? Mistake

Response: Corrected.

Line 260 mnagmenet??? Mistake

Response: Corrected.

 Table 6. An empty column in the table?....In addition, questions arise about the meaning of correlations in this table, since they are all negative, although the processes of mineralization and sequestration are of opposite directions in their essence.

Response:  Sorry for the typing mistake. Now they have been corrected.

In the Conclusions section, I would suggest specifying in more details the influence of legumes and specific grasses combinations on carbon storage, enzymatic stoichiometry, etc.

Response: We have specified the influence of legumes and specific grasses combinations on carbon storage, enzymatic stoichiometry

Round 2

Reviewer 2 Report

Comments and Suggestions for Authors

The paper has been improved.

Author Response

Thanks for your consideration and positive remarks.

Reviewer 3 Report

Comments and Suggestions for Authors

The authors did not correct the main drawback, namely, an incorrect understanding of the scientific basis of the experiment and the interpretation of the results. If they do not do this in the next version, the work should be rejected so as not to discredit the scientific level of the journal. 

Instead of meaningfully correcting this deficiency, they repeated the same phrases THREE times about how their task was to compare fertilized and unfertilized pasture options. The presence of such a task cannot allow one to violate one of the most basic principles of scientific analysis. Moreover, repeating these phrases three times instead of answering in essence can be regarded as disrespect for the work of the reviewer, which gave me the moral right to simply write that the work should be rejected.

However, given the large amount of data collection work done by the authors, I am making another attempt to indicate to the authors what needs to be corrected in the article.

Based on a comparison of the fertilized experimental plots SG/SGL and unfertilized natural pastures NG, authors cannot draw conclusions that the found changes in the parameters of the experimental plots SG/SGL and natural pastures NG are due to a different number of legume species. All these changes can also be caused by fertilizer. It is well known that fertilizer has a wide range of effects on both species diversity and ecosystem functions.

The necessary corrections must be made to the following lines of the revised version of the manuscript (PDF file attached):

24, 29, 216, 345, 347, 371, 437-438, titles of tables 1-5 (lines 220, 240, 261, 314, 352).

Also take into account two small notes on the lines 130-133 and140.

If the authors for some reason do not make the above basic corrections, the manuscript, in my opinion, should be rejected

Author Response

We beg to differ with your observation, specifically omitting two treatments, namely Natural pasture (NG) and uncultivated fallow (F). We are happy to explain the reason.

In any scientific experiment of natural science, we must have a control/reference to compare the supremacy or inferiority of intervention. In compliance with this principle, we compared the impact of legume diversification with standard package of practices (including fertilizer application, irrigation, etc) on soil carbon storage, enzymatic stoichiometry, and ecosystem functions in the Indian Himalayan legume-diversified pastures. To test the supremacy or inferiority of legume diversification with standard package of practices, we compare them with natural pasture (NG) and uncultivated fallow (F). Now, if we delete these two treatments, namely natural pasture (NG) and uncultivated fallow (F), we will lose our reference point. This will violate the simple principle of experimentation of natural science. However, as pointed you, we have added fertilization as a possible reason for greater biomass productivity and carbon accumulation in relevant subsections of Results and Discussions. If we delete treatments, namely Natural pasture (NG) and uncultivated fallow (F), then we cannot draw any conclusion and recommend legume diversification, if we do not have comparison with sole NG. 

We have also incorporated all other suggestion as mentioned by you in the comment section.

We look forward to your wisdom, understanding and considerations!